# The Therapeutic Potential of Pyroptosis in Melanoma

**DOI:** 10.3390/ijms24021285

**Published:** 2023-01-09

**Authors:** Nadia Zaffaroni, Giovanni L. Beretta

**Affiliations:** Molecular Pharmacology Unit, Department of Experimental Oncology, Fondazione IRCCS Istituto Nazionale dei Tumori, 20133 Milan, Italy

**Keywords:** pyroptosis, melanoma, gasdermin, drug combinations, gene signature

## Abstract

Pyroptosis is a programmed cell death characterized by the rupture of the plasma membranes and release of cellular content leading to inflammatory reaction. Four cellular mechanisms inducing pyroptosis have been reported thus far, including the (i) caspase 1-mediated canonical, (ii) caspase 4/5/11-mediated non-canonical, (iii) caspase 3/8-mediated and (iv) caspase-independent pathways. Although discovered as a defense mechanism protecting cells from infections of intracellular pathogens, pyroptosis plays roles in tumor initiation, progression and metastasis of tumors, as well as in treatment response to antitumor drugs and, consequently, patient outcome. Pyroptosis induction following antitumor therapies has been reported in several tumor types, including lung, colorectal and gastric cancer, hepatocellular carcinoma and melanoma. This review provides an overview of the cellular pathways of pyroptosis and discusses the therapeutic potential of pyroptosis induction in cancer, particularly in melanoma.

## 1. Introduction

Cancer therapies are mainly based on the induction of cell death. The mechanisms of cell death include accidental cell death (ACD) and programmed cell death (PCD) [1]. ACD is a non-controlled process caused by extreme extracellular stress conditions (e.g., chemical, pressure, temperature and osmotic pressure) that the cell is unable to control to return to physiologic status, which leads to cell membrane rupture and cytoplasm release. Though the cytoplasm release is a hallmark of ACD, it occurs during PCD as well [2,3]. In contrast to ACD, PCD is a controlled cell suicide regulated by cellular mechanisms involving biochemical and immunological pathways that remove harmful or abnormal cells and govern physiological and pathological processes. In addition to apoptosis, which is the most well-known mechanism, PCD includes necroptosis, ferroptosis, autophagy and pyroptosis (PYR) [4,5,6,7,8,9,10,11]. Recent studies report a new PCD mechanism, PANoptosis, which is based on the formation of the PANoptosome, a molecular complex composed of molecules governing apoptosis, necroptosis and PYR [12,13]. The diverse PCD mechanisms are classified on the basis of gene expression, biochemical and cellular morphological properties as well as different effector molecules representing potential selective biomarkers and targets useful for fighting cancer [14,15]. In this regard, the discovery of drugs that stimulate apoptosis has attracted attention and research investment [16]. Though productive, this strategy has often failed due to the emergence of several drawbacks. Among the inconveniences, the development of tumors resistant to apoptosis following drug treatment is crucial [17]. Thus, the discovery of drugs that induce PCD unrelated to apoptosis is an interesting approach to anticancer therapy. In this context, it is intriguing to consider the compounds that induce PYR in cancer, particularly in melanoma [18,19,20,21,22,23,24,25,26,27,28,29,30,31,32].

Melanoma is responsible for the majority of deaths from skin cancers. In 2020 the number of new cases and deaths worldwide for cutaneous melanoma skin compared to all cancers were 1.7% and 0.6%, respectively [33]. Though a reduction in mortality is reported, likely dependent on success in mass-media prevention campaigns as well as on the improvement of the medical management of this pathology [34], an early diagnosis is fundamental as late-diagnosed melanoma still remains an incurable disease. Mutations of the BRAF gene, among which V600E is the most frequent (around 90% of cases), result in constitutive activation of the RAS-RAF-MEK-ERK axis, which accounts for disease aggressiveness. The clinical introduction of the kinase inhibitor vemurafenib, which is selective for BRAFV600E-mutated melanoma, proved efficacious in patients suffering from metastatic melanoma [35,36]. Due to cellular mechanisms reactivating the MAPK pathway, patients treated with BRAF inhibitors (BRAFi) develop resistance after a few months [37]. Despite the increased effectiveness observed in BRAFi-resistant patients treated with the combination of BRAFi and MEK-inhibitors (MEKi), the achievement of persistent cures is limited by the emergence of drug resistance towards the drug combination as well [38,39,40]. Although BRAFi/MEKi-resistant patients can benefit from different therapeutic opportunities, such as immune checkpoint inhibitors (ICIs), strategies aimed at overcoming drug resistance remain an urgent need.

This review provides an overview of the molecular mechanisms of PYR and discusses the possible implications of PYR induction for melanoma therapy.

## 2. Molecular Mechanisms of Pyroptosis: General Overview

PYR, also known as inflammatory necrosis, is a PCD that antagonizes microorganism infections and dangerous intracellular signals. PYR was first observed by Friedlander in 1986 in murine macrophages showing unconventional morphology following exposure to anthrax lethal toxin [41]. The definition of PYR dates back to 2001 and refers to a peculiar and rapid programmed necrosis implicating caspase 1 activation and inflammation induced by Salmonella in macrophages [42]. Later, caspases 3, 7 and 8 and proteins belonging to the gasdermin (GSDM) family were reported to play a role in PYR induction as well [43,44]. In 2018, PYR was defined by the Nomenclature Committee on Cell Death as an inflammatory PCD mediated by GSDM proteins [1]. The proteolysis of GSDM is the crucial step for PYR induction. The site-specific proteolysis of GSDM induces the shedding of the pore-forming N-terminus domain (NT-GSDM), which oligomerizes and forms membrane pores by interacting with phosphatidylinositol phosphates, phosphatidylserine and cardiolipin in the cell membrane. Pore/channel formation in the plasma membrane stimulates the cytoplasm release into the extracellular compartment, inducing cell swelling, inflammation and leading to PYR [45].

Pyroptotic cells display morphological features similar to apoptosis and necroptosis. Similar to apoptosis, these cells show membrane blebbing, DNA fragmentation and chromatin condensation, activation of caspases 3/6/8/9 as well as poly (ADP-ribose) polymerase cleavage [1]. Conversely, DNA fragmentation not associated with chromatin condensation, intact nuclei, cell swelling due to the rupture of the plasma membrane, osmotic lysis and induction of inflammation are properties of PYR that are in common with necroptosis. Peculiar characteristics of PYR are the activation of caspases 1/4/5/11, GSDM cleavage and the formation of the pyroptotic bodies [46]. The oligomerization of specific peptide-forming pores and the translocation of these aggregates on the plasma membrane leading to cellular permeabilization are two pivotal steps in common between necroptosis and PYR. Though similar, the cellular processes guiding the formation of the membrane pores implicate the activation of diverse molecules; mixed-lineage kinase domain-like protein (MLKL) in necroptosis and GSDM in PYR [46,47].

Pattern recognition receptors (PRRs) are membrane receptors that play an important role in the body’s first defense against pathogens. PRRs include Toll-like receptors (TLRs) and NOD-like receptors (NLRs), which recognize pathogen-associated molecular patterns (PAMPs) and damage-associated molecular patterns (DAMPs). PAMPs are molecules of pathogenic microorganisms and include membrane components, lipoprotein, surface glycoprotein and nucleic acid, while DAMPs are endogenous molecules derived from the damaged host-infected cells [48,49,50,51]. The activation of TLRs favors the release of inflammatory cytokines such as tumor necrosis factor (TNF), IL 6 and IL 8. Conversely, dangerous signals hitting host cells invaded by intracellular pathogens are recognized by NLRs and lead to the production of inflammatory cytokines different from those released by the activation of TLRs. These cytokines favor local inflammation accompanied by the recruitment and priming of immune cells, allowing the pyroptotic death of the host cells and the elimination of pathogens from the host [52].

Overall, the cellular mechanisms of PYR can be divided into caspase-dependent, which includes canonical and non-canonical pathways, and caspase-independent pathways. The family of caspases contains pro-apoptotic caspases (e.g., initiators—caspase 8, 9, 10; executors—caspase 3, 6, 7) and pro-inflammatory caspases (caspases 1, 4 and 5) [53,54]. This notion is not rigid and, although PYR is regulated by pro-inflammatory caspases, subtle cellular regulations promote the switch from specific caspase-mediated apoptosis to PYR. For example, caspase 3-mediated apoptosis is converted to PYR by TNF or chemotherapeutic agents [55]. Similarly, YopJ, the effector of Yersinia pestis, induces caspase 8-mediated PYR by inhibiting TAK1-IκB kinase signalling, which governs caspase 8-dependent apoptosis [56].

Four cellular mechanisms of PYR induction have been reported thus far (Figure 1 and Figure 2), including the (i) caspase 1-mediated canonical, (ii) caspase 4/5/11-mediated non-canonical, (iii) caspase 3/8-mediated and (iv) caspase-independent pathways. These mechanisms comprise four phases, including (i) signal capture, (ii) transmission of the signal, (iii) activation of PYR executors and (iv) PYR.

PRR engagement activates signals that stimulate the caspase 1-mediated canonical pathway. Conversely, the activation of the other 3 pathways depends on diverse cellular mechanisms, including secretion/endocytosis (as in the case of the YopJ factor and lipopolysaccharides, LPS) and perforin released by natural killer (NK) cells and cytotoxic T lymphocytes (CTLs) for the delivery of granzyme (GZM) [57,58,59].

Caspase 1-mediated canonical pathway. Signals that favor the formation of the inflammasome and the activation of caspase 1 stimulate PYR (Figure 1). The inflammasome contains apoptosis-associated speck-like protein (ASC), NLR, and procaspase 1. PAMPs as well as DAMPs of host or tumor cells are recognized by NLR that, following activation, interacts with the adaptor protein ASC, which contains the caspase activation and recruitment domain (CARD) [60]. The conformational changes induced by the NLR/ASC interaction stimulate the recruitment of procaspase 1, leading to the organization of a complete inflammasome, which favors procaspase 1 cleavage. Activated caspase 1 stimulates the proteolysis of GSDM D (at the Asp270 residue), releasing NT-GSDM D and leading to alterations of plasma membrane integrity, cell swelling and PYR [61,62]. The activation of caspase 1 also stimulates the proteolysis of pro-IL1β and pro-IL18, allowing the release of activated IL1β and IL18 [45]. PYR can be counteracted by the calcium influx mediated by the endosomal sorting complexes required for the transport (ESCRT) machinery, which triggers membrane repair by removing the NT-GSDM D-mediated pores [63]. In this regard, the Ca^2+^ chelator BAPTA-AM encapsulated into biodegradable dextran nanoparticles (NPs) showed efficacy in inhibiting the ESCRT machinery and in potentiating PYR induction in in vivo models of metastatic breast and ovarian cancers and melanoma [64].

Caspase 4/5/11-mediated non-canonical pathway. The non-canonical pathway involves caspase 11 in mouse and its human homologs, caspase 4 and 5 (Figure 1). The CARD of caspase 4/5/11 recognizes LPS and induces oligomerization and proteolytic-mediated caspase self-activation. Activated caspase leads to the release of NT-GSDM D, the formation of membrane pores, and PYR [43,44,65,66]. Caspase 4/5/11 are unable to directly stimulate the activation of pro-IL18 and pro-IL1β, and the inflammasome governing this non-canonical pathway is the NOD-like receptor family pyrin domain containing 3 (NLRP3) inflammasome. The release of IL1β and IL18 occurs following the activation of NLRP3 and is mediated by potassium efflux, which is stimulated by the formation of caspase 4/5/11-mediated NT-GSDM D pores [67,68,69]. An alternative non-canonical pathway involves the caspase 11-mediated cleavage of the pannexin-1 (PANX1) channel. This process stimulates the release of ATP in the extracellular compartment, which activates the P2X ligand-gated ion channel (P2 × 7) receptors, in turn leading to PYR [70].

Caspase 3/8-mediated pathway. Although recognized as crucial factors for the induction of necroptosis and apoptosis, caspase 3 and 8 are implicated in PYR as well (Figure 2 [71]). Following caspase 3 activation, GSDM E is cleaved (at the Asp270 residue) and releases NT-GSDM E aggregates to form membrane pores, leading to the disruption of plasma membrane integrity and in turn to PYR. Of note, PYR is counteracted by caspase 3 cleavage of GSDM E at a different cleavage site (Asp87 instead of Asp270), allowing the production of shorter NT-GSDM E peptides unable to form functional membrane pores [72]. PYR is also induced by the proteolytic activation of GSDM C, D and E mediated by caspase 8 activation [73,74]. This scenario renders caspase 8 the key switch that decides the cellular fate among 3 main PCD pathways, apoptosis, necroptosis and PYR [75].

Caspase-independent pathway. In addition to caspase-mediated induction, PYR is stimulated by other enzymes that activate GSDM, including GZM (Figure 2). GZMs are serine proteases released by CTLs and NK cells that cleave specific protein substrates of target cells leading to PCD [59]. This family of proteases comprises 5 enzymes endowed with proteolytic activity toward caspases and promoting apoptosis. Recently, GZM A and GZM B were reported to favor PYR [76,77]. GZM B induces PYR by cleaving GSDM B. In the same way, GZM A favors PYR by proteolyzing GSDM E in the same site recognized by caspase 3.

## 3. Pyroptosis and Cancer

In addition to its role in protecting cells from pathogen infections, PYR is implicated in cancer. Due to the engagement of immune cells that stimulate inflammation, PYR can promote tumor inhibition [78,79]. The anticancer potential of PYR relies on the release of the cellular content, which includes proinflammatory cytokines (e.g., mature IL1 and IL18), tumor antigens, ATP and DAMPs. This behavior stimulates adaptive immunity and antigen presentation as well as TLR activation [80,81]. Released ATP stimulates the activation of P2 × 7 receptors and, in this way, the formation of additional membrane pores that enhance inflammation. In addition to the formation of new pores in pyroptotic cells, the extracellular ATP activates P2 × 7 receptors of dendritic cells (DCs), stimulating the recruitment of CTLs and antitumor immunity [82,83,84,85,86]. Following plasma membrane rupture, the release into the extracellular compartment of high-mobility group box-1 (HMGB1) occurs [87,88]. The oxidative tumor microenvironment (TME) favors the oxidation of HMGB1 leading to the formation of diverse oxidized forms. Among these, disulfide-HMGB1 favors the release of cytokines and stimulates an anticancer proinflammatory environment [89,90]. Other oxidized forms of HMGB1 completely abolish its proinflammatory activity. These oxidized forms of HMGB1 are typically recognized during apoptosis and are responsible for the immune tolerance associated with apoptotic PCD [91].

PYR induction has been reported in non-small cell lung cancer (NSCLC) treated with simvastatin and polyphyllin VI [18,19]. NLRP3 inflammasome and caspase 1 activation by simvastatin stimulates PYR through the canonical pathway that inhibits NSCLC growth. Similarly, polyphyllin VI, a compound isolated from Trillium tschonoskii maxim, reduces NSCLC growth by inducing PYR through the activation of caspase 1 and GSDM D proteolysis mediated by the NLRP3 inflammasome. Berberine and sorafenib induce PYR in hepatocellular carcinoma (HCC) [20,21]. Berberine stimulates PYR in HepG2 cells by promoting caspase 1 activation, which reduces cell proliferation, migration and HCC growth in vivo. In addition to its direct action on cancer cells and angiogenesis, sorafenib induces PYR in macrophages, favoring the release of proinflammatory cytokines and activation of NK cells and leading to reduced HCC growth in vivo. PYR induced by 5-aza-2-deoxycytidine (DAC) and lobaplatin is reported in colorectal cancer (CRC) models (including, LAS174T, LoVo, HCT116 and HT29 cell lines) [22,23]. DAC up-regulates the NLRP1 inflammasome in CRC. After exposure to DAC, NLRP1 activation increases in CRC and this feature stimulates tumor inhibition in vivo via PYR induction. A reduced cell viability dependent on PYR induction has been reported in lobaplatin-exposed HT29 and HCT116 cells. Following lobaplatin treatment, these cells undergo PYR stimulated by the caspase 3-mediated cleavage of GSDM E. Similarly, in gastric cancer (GC) the levels of GSDM E are implemented by the exposure to 5-fluorouracil (5-FU). GC cells treated with 5-FU show caspase 3-mediated activation of GSDM E and PYR induction [24].

### 3.1. Drug Combinations That Induce Pyroptosis in Melanoma

Surgery, conventional and targeted chemotherapy, radiotherapy and immunotherapy are used to manage patients suffering from melanoma. Radiotherapeutic and pharmacological approaches kill tumor cells and counteract tumor proliferation primarily by inducing apoptosis. Despite the positive results achieved, late-diagnosed melanoma still remains incurable, and this implies that new medical strategies are urgently needed. In this scenario, drug combinations performed in preclinical models based on conventional anticancer drugs demonstrate antitumor efficacy in melanoma via PYR induction, Figure 3 and Table 1.

The combination BRAFi/MEKi is FDA-approved for the treatment of BRAF-mutated melanoma patients. Erkes and co-workers demonstrated that a proficient immune system is required for the antitumor efficacy of the combination of PLX4720 and PD0325901 [25]. The study showed that T cell accumulation/activation at the tumor site activated caspase 3 and GSDM E cleavage, favoring the release of HMGB1 and PYR induction. Consistent with this notion, cells lacking GSDM E were insensitive to the drug combination and showed defective HMGB1 release, reduced tumor-associated T cell infiltrates, and frequent tumor re-growth after drug removal. Since resistance to BRAFi/MEKi is associated with poor intratumoral T cell accumulation/activation and reduced PYR induction, the combination of BRAFi/MEKi with drugs that stimulate PYR represents a potential salvage therapy in such patients.

Another drug combination strategy to counteract BRAFi/MEKi resistance was proposed by Cai et al. [26]. An increased sensitivity to the MEKi trametinib was reported in melanoma cell lines, including WM1361A, WM1633, SK-MEL-30 and SK-MEL-173, in which phosphoinositide-dependent kinase-1 (PDPK1) level is reduced by specific siRNA. PDPK1 acts downstream of PI3K and activates oncogenic pathways, including AKT, PKC, p70S6K, SGK, PLCg1, and Plk/cMyc, that favor tumor growth. In this study, the small molecule GSK2334470 was used to inhibit PDPK1 in vivo. Compared to single drug treatment, the combined exposure to trametinib and GSK2334470 significantly reduced tumor growth and increased survival of SK-MEL-30 xenograft-bearing mice. A deeper investigation demonstrated that the GSK2334470/trametinib combination suppressed tumor growth by inducing caspase 3-mediated activation of GSDM E, in turn leading to PYR. Additional features reflecting PYR were the typical morphological changes and the release of HMGB1. The contribution of the immune system to the efficacy of GSK2334470/trametinib was also evaluated using immunocompetent as well as immunocompromised allograft mouse models. Compared to immunocompromised mice, immunocompetent animals showed higher levels of intratumoral CD8+ T cells with increased tumor growth inhibition and prolonged survival.

The study by Ahmed and colleagues was performed using primary cell lines collected before and after patients’ exposure to BRAFi [27]. The combination of temozolomide and chloroquine was tested on the panel of BRAFi-sensitive and -resistant cells and no perfect match in terms of cell response was observed. Sensitive and resistant CM143 cells were selected for further investigations and, compared to BRAFi-sensitive CM143 cells, the drug combination was more active in reducing the proliferation of resistant cells. In addition, an increased release of IL1β was reported in BRAFi-resistant cells. The exposure to the drug combination induced caspase 3 activation and GSDM E/D cleavage, thus suggesting PYR induction. Of note, the combination showed better antitumor activity in xenograft BRAFi-resistant CM143-bearing mice with respect to singly administered temozolomide.

Conventional drug-based therapies are severely limited due to the lack of specificity towards cancer cells and high toxicity to healthy tissues. Although the combination of metformin (MET) and doxorubicin (DOX) is effective in treating numerous cancers, including melanoma, clinical limitations are reported, including the short half-life and poor bioavailability of MET, the side effects occurring at high doses, and the differences in chemical properties of the two drugs (e.g., DOX hydrophobicity and MET hydrophilicity). These features lead to reduced effective co-accumulation of the drugs into the tumor. To overcome this drawback, Song and colleagues proposed the delivery of the combination MET/DOX via a polymeric pH-sensitive, tumor-targeting, and biocompatible NPs [28]. These NPs are composed of sodium alginate and contain cholesterol and folic acid (FCA), two essential but insufficient substrates for melanoma growth. Empty NPs were safe in vitro and in vivo in C57BL6/J mice. Treated animals showed only modest effects on body weight and no significant histological lesions as well as serological alterations. Moreover, NPs efficiently accumulated in A375 melanoma cells via clathrin-mediated FCA uptake. Specific tumor targeting favored by FCA was reported in vivo in A375 tumor-bearing mice as well. MET and DOX were efficiently loaded on NPs. MET/DOX-loaded NPs were more active than free MET and DOX administered alone or in combination in A375 and SK-MEL-28 cells and in A375 tumor-bearing mice. Deeper investigations into cell death mechanisms indicated that loaded NPs induced cell death via PANoptosis both in vitro and in vivo. Indeed, biochemical markers of apoptosis (cleaved caspase 7), necroptosis (MLKL) and PYR (GSDM D) were observed following exposure to NPs.

### 3.2. Pyroptosis-Associated Gene Signatures in Melanoma

Several investigations have focused on the construction of PYR-associated gene signatures for predicting melanoma patient outcomes. These studies also predict sensitivity to antitumor drugs and allow the identification of potential targets for novel clinical interventions (Table 2). The studies collected data from 3 databases available online, including The Cancer Genome Atlas (TCGA) and Gene Expression Omnibus (GEO) for melanoma patients, and Genotype-Tissue Expression (GTEx) for normal skin subjects.

By analyzing gene expression profiles of normal skin and melanoma cells, Li and colleagues reported a set of 5 key prognostic, differentially expressed PYR-associated genes (GSDM A, GSDM C, IL18, NLRP6 and AIM2) [92]. The risk score allowed the classification of TCGA patients into high-risk and low-risk groups. A difference in overall survival (OS) was observed between the two groups, with patients belonging to the high-risk group showing higher mortality than those in the low-risk group. TIMER database analysis indicated that the signature shows a correlation with infiltration of immune cells, including CD8+ T and CD4+ T cells, neutrophils and DCs. By applying the pRRophetic algorithm, the study also predicted differences in sensitivity to conventional chemotherapeutics (paclitaxel, docetaxel and cisplatin), targeted therapies (the kinase inhibitors sorafenib and PD0325901) and ICIs (targeting the immune checkpoint genes PD1, PD-L1, CTLA4, LAG3, or VSIR) in the two risk groups. The analysis predicted greater sensitivity to paclitaxel, sorafenib and PD0325901, and lesser sensitivity to cisplatin and docetaxel for low-risk patients compared to high-risk patients. Moreover, since PD1, PD-L1, CTLA4, LAG3, and VSIR were highly expressed in the high-risk group, these patients were expected to be more sensitive to inhibitors of these immune detection targets. The signature was validated by comparing the expression of the genes in normal skin HaCaT cells and melanoma A375, HS294T and M14 cell lines. Compared to HaCaT, melanoma cells showed lower levels of GSDM A, GSDM C and IL18 and higher levels of NLRP6. Increased expression of AIM2 was reported for A375 and HS294T compared to healthy cells.

An eight-genes signature comprising genes related to inflammation and PYR has been proposed by Xu et al. [93]. The study considered the melanoma TCGA and normal skin GTEx data for the construction of the training cohort. After univariable Cox and least absolute shrinkage and selection operator (LASSO) regression analysis, and multivariable Cox regression analysis, a prognostic signature (TLR1, CCL8, EMP3, IFNGR2, CCL25, IL15, RTP4 and NLRP6) was constructed. Such a signature allowed the stratification of the patients of the training cohort into high- and low-risk groups, with an OS rate of the high-risk-group significantly lower than that of the low-risk group. Gene-set enrichment analysis (GSEA) showed that the TME of the low-risk group is enriched in immune cells, including infiltrating CD8+ T and T helper cells as well as tumor infiltrating lymphocytes. Compared to high-risk ones, low-risk patients, whose tumors express PD1 or CTLA4, better responded to ICIs. The drug sensitivity analysis, which considered 17 targeted drugs (e.g., afatinib, sorafenib and refametinib) and 12 conventional therapeutics (e.g., docetaxel, rapamycin, cisplatin), showed different sensitivity for the two groups. Moreover, the signature was validated by immunohistochemical data extracted from the Human Protein Atlas and by qRT-PCR analysis carried out on normal human immortalized HaCaT keratinocytes, human skin PIG1 melanocyte and melanoma A375, SK-MEL-28 cell lines. Although the study underscores that PYR and inflammation responses predict the prognosis and immunotherapy response of patients suffering from melanoma, the authors themselves evidenced two major limitations, including (i) the lack of an independent patient cohort to better validate the prognostic power of the model and (ii) the lack of validation resulting from the analysis of clinical samples.

Another signature based on PYR-related genes was reported by Wang and colleagues [94]. The signature demonstrates differences in TME composition and predicts prognosis as well as response to immunotherapy of melanoma-suffering patients. By applying the GEPIA2 online software to the expression levels of GSDM and inflammasome-related genes extracted from TCGA and GEO (for melanoma patients) and GTEx (for normal skin subjects) databases, a gene signature associated with PYR was defined, including AIM2, GSDM C, GSDM D, IL18, NLRP6 and PRKACA). The signature allowed the construction of a risk model that stratifies patients into high- and low-risk groups. The better prognosis observed for the low-risk group associated with higher expression of PYR-related genes; higher proportion of infiltrating memory B cells, CD8+ T cells, activated memory CD4+ T cells, Tregs and M1 macrophages; and with a lower proportion of M2 and M0 macrophages, as well as resting mast cells. Compared to high-risk, the low-risk group also showed higher expression of immunoinhibitory genes and MHC-related genes and more immunosuppressive Tregs. These features denote that the low-risk group has an immune-proficient TME, which favors immune cell infiltration and sensitivity to immunotherapy (e.g., against PD1 and CTLA4). Receiver operating characteristic (ROC) analysis indicated that this prognostic risk model effectively predicted patient prognosis.

The study by Niu and colleagues collected data of melanoma patients from TCGA database for the construction of a training cohort and an internal validation cohort, and data of normal skin subject from the GTEx database [95]. Moreover, data from the GEO database (GSE65904) worked as an external validation cohort. To identify prognostic genes and conceive a risk score, gene expression levels collected from the databases were analyzed by Cox and LASSO regressions. The resulting four-gene signature, including GSDM A, GSDM C, AIM2 and NOD2, and risk score allowed the classification of the patients into high-risk and low-risk groups. The prognostic model predicted significant differences in OS for the two groups which was corroborated by internal and external validation cohorts. Gene Ontology (GO) and Kyoto Encyclopedia of Genes and Genomes (KEGG) analyses clearly demonstrated differences between the two groups, mainly involving immune-related signaling pathways. Compared to high-risk, the low-risk-group showed an up-regulation of all the immune-related pathways and higher levels of key antitumor infiltrating immune cells. Additionally, PD1, PD-L1, PD-L2 and CTLA4 were highly expressed in low-risk patients, who responded better to ICIs (PD1 and CTLA4 blockers).

The expression levels of a set of PYR-related genes from melanoma patients (TCGA and GSE65904) and healthy individuals (GTEx) were considered by Meng and co-workers for the definition of a signature including 12 differentially expressed genes (AIM2, IL1B, NLRC4, NLRP3, NLRP6, NLRP7, TNF, ELANE, GSDM A, GSDM B, GSDM C, NLRP1) [96]. The matching of the gene expression levels with OS information stratified the patients into 3 clusters. One of these clusters highly expressed the genes of the signature and showed enrichment in pathways related to immune cell activation (apoptosis, chemokine, NK cell-mediated cytotoxicity, T-cell receptor, and B-cell receptor-related signaling pathways) as well as enrichment in immune cell content (CD4+, CD8+ T cells, and immature B cells). The defined PYR score represents an independent prognostic factor. A high PYR score reflects patient survival advantage, an immune-proficient infiltrated TME, and associates with high levels of PD1, PD-L1 and CTLA4, in turn indicating a superior therapeutic benefit by ICIs (anti-CTLA4 and anti-PD1).

Another PYR-related gene signature is proposed by Wu et al. [97]. Gene profiles and clinical data of melanoma patients from TCGA and GEO matched with that of normal subjects allowed the identification of differentially expressed genes associated with PYR. Univariate Cox and LASSO analyses defined a PYR-related risk gene signature, including GSDM C, GZMA, AIM2 and PD-L1 that is associated with prognosis. According to the risk score, patients are divided into low- and high-risk groups, with patients in the high-risk group showing lower OS. Moreover, the Kaplan–Meier (K–M) analysis confirmed the association of the signature with prognosis. The analysis of the relationship between immune status and the risk signature indicates that all the immune cell subpopulations were reduced in the high-risk group. In addition, GSEA confirmed that several enriched pathways were associated with immunity, including NK cell-mediated cytotoxicity as well as T-cell receptor and TLR signaling pathways. The expression of the four genes of the signature significantly correlated with the sensitivity towards several antitumor drugs, including nelarabine, dexamethasone decadron, fluphenazine, arsenic trioxide, procarbazine, olaparib, fludarabine, simvastatin, cyclophosphamide, and asparaginase.

A nine-gene signature was constructed by Ju and colleagues by analyzing the expression profiles of 20 genes playing a central role in PYR induction that were downloaded from TCGA and GTEx databases [98]. The signature (NLRP9, DHX9, CASP3, NLRC4, AIM2, NLRP3, IL1B, GSDM E and GSDM D) demonstrates powerfulness as a melanoma diagnosis tool, and separated with high accuracy primary melanoma patients from subjects suffering from common nevi belonging to an independent (GSE98394) dataset. According to the risk score of the prognostic model, patients from TCGA and from a validation cohort were divided into low- and high-risk groups. In comparison with patients of the low-risk group, a shorter lifespan was observed for patients of the high-risk group, as defined by K–M survival analysis. The functional GSEA and the estimation of immune cell components by CIBERSORT revealed a close association with activation of pathways of the immune response, as well as a peculiar proportion of immune cell components in the TME reported for the low-risk group.

Lou and colleagues analyzed mRNA levels of 17 PYR-associated genes in 17 types of cancer (TCGA) and reported an increased expression of these genes in tumor-suffering patients with high-immunocompetence (e.g., TME immune infiltration and immune activation) [99]. The prognostic potential of the signature was confirmed in an additional 33 cancer types and the K–M analysis confirmed that the signature predicted survival in melanoma patients. In order to improve the model accuracy and decrease model overfitting, LASSO analysis was applied and a more accurate risk model based on the expression of 6 PYR-associated genes (CASP5, NEK7, AIM2, CASP1, NLRC4, GSDM D) was defined. The risk score further allowed the stratification of melanoma patients into high-risk and low-risk groups, with a survival benefit for the low-risk group. The elevated level of PYR-associated genes predicted better survival rate (ROC analysis) and strongly associated with clinicopathological features of the patients. A deeper investigation carried out on an independent set of melanoma patients treated or not with immunotherapy correlated the signature with the response to anti-PD1 therapy. Protein levels of CASP1, PYCARD, and CASP4 in patients responding to therapy were significantly higher than that observed in the non-responder group. These results were confirmed in A375 cells transfected with plasmid allowing PD1 overexpression that showed increased CASP1, CASP4 and PYCARD levels.

Shi and colleagues constructed a 3 PYR-related genes signature, including BST2, GBP5 and AIM2, analyzing the expression profiles from TCGA and GTEx platforms [100]. Data from the GEO database (GSE65904) was used as a validation cohort. The risk model stratified the TCGA and GSE65904 patients into high- and low-risk groups. The two groups showed different OS, with lower OS for high-risk compared to low-risk patients. While ROC analysis indicated a moderate predictive accuracy, the risk model had a higher predictive power in comparison to clinical characteristics. Nomograms defined on the basis of the risk model showed enhanced discriminatory abilities for melanoma patient outcome.

In the study by Wu and colleagues, a PYR-based model was constructed by analyzing RNA sequencing data and clinical information of melanoma patients from four immunotherapy databases, including Gide (patients receiving anti-PD1 or the combination anti-PD1 and anti-CTLA4), Lauss (patients treated with adoptive T-cell therapy), Liu (patients treated with anti-PD1) and Nathanson (patients treated with anti-CTLA4) [101]. Gide worked as the training cohort and the others as validation cohorts. Moreover, data from the TCGA-SKCM database was considered as a control cohort of melanoma patients not receiving immunotherapy. The PYR-based model was constructed by analyzing a gene set of 33 PYR-related genes that, after LASSO regression analysis, allowed the identification of four genes (CASP5, NLRP6, NLRP7, PYCARD) significantly associated with immunotherapy. Following the application of four machine-learning methods, a model (e.g., PYR score) for predicting clinical benefits from immunotherapy was proposed. The model predicted durable clinical benefits of immunotherapy and this finding was confirmed by the ROC analysis as well. The K–M analysis showed that, compared to low PYR scores, high PYR scores were associated with favorable OS and progression-free survival. These findings were not observed for the TCGA-SKCM cohort, which included melanoma patients not receiving immunotherapy, supporting the specificity of the score. Moreover, the model was only predictive for melanoma patients treated with immunotherapy and not for subjects suffering from other cancer types receiving immunotherapy, including metastatic GC and advanced clear-cell renal cell carcinoma. The molecular analysis performed applying GO and KEGG on a set of differentially expressed genes in tumors showing different PYR scores followed by a GSEA indicated that high PYR scores were associated with an immune-inflamed phenotype, including enrichment of immunostimulatory pathways, increased level of tumor-infiltrating lymphocytes, upregulation of immune effectors, and activation of the antitumor immune response. Moreover, the application of the CIBERSORT algorithm to estimate the relative proportion of tumor-infiltrating immune cells in TME showed that high PYR scores were associated with elevated infiltration level of CD8+ T cells, activated memory CD4+ T cells, polarized M1 and M2 macrophages and plasma cells. Conversely, tumors with low PYR scores contained more resting immune cells, including naïve CD4+ T cells and M0 macrophages.

Wang and colleagues analyzed the mRNA expression profiles of untreated and BRAFi-treated melanoma cells from 3 datasets (GSE42872, GSE52882 and GSE106321) registered into the GEO platform [102]. Differentially expressed genes were analyzed with GO and KEGG and an enrichment in the Jak-STAT signaling pathway, with a notable increased expression of IRF9 and STAT2 in the treated samples, was reported. These results were validated in A375 and SK-MEL-28 cell lines in vitro and in vivo. The overexpression of IRF9 or STAT2 results in reduced sensitivity to vemurafenib. Conversely, IRF9 or STAT2 knockdown increases the sensitivity towards the BRAFi. Similarly, in vivo results show that the overexpression of IRF9 or STAT2 delays vemurafenib-induced tumor regression, whereas knockdown of IRF9 or STAT2 potentiates tumor growth inhibition. Specific bioinformatics tools predict an interaction of STAT2 with the GSDM E promoter, a finding validated by the chromatin immunoprecipitation assay. This interaction reduces GSDM E expression and in turn PYR. This implies that drugs inhibiting the IRF9-STAT2 signaling upregulate GSDM E-mediated PYR and overcome adaptive resistance induced by vemurafenib exposure.

In the study by Xie J. and colleagues, a signature based on lncRNAs related to PYR was proposed as a prognostic tool [103]. A set of lncRNAs associated with PYR was selected and their expression values in melanoma patients were downloaded from TCGA. Through Cox and LASSO regressions, a prognostic model enclosing 9 lncRNAs, including AL121603.2, AC107464.2, AC245128.3, AC092171.5, AC242842.1, IRF2-DT, HLA-DQB1-AS1, AC004585.1 and LINC00582, was constructed. The calculated risk score allowed the patients’ stratification into high-risk or low-risk groups. Compared to the low-risk group, a reduced survival rate was recognized for patients in the high-risk group. The potency and the accuracy of the model was confirmed by ROC and calibration curves. Moreover, patients in the low-risk group had higher levels of immune cells in the TME and expressed higher levels of immune checkpoint-related and M6A methylation-related genes. The evaluation of the drug sensitivity via dedicated software predicted a major sensitivity to bexarotene, bryostatin and docetaxel for the high-risk group, and to bortezomib, bosutinib and camptothecin for the low-risk group. GSEA analysis showed enrichment of genes related to extracellular matrix-receptor interactions, cancer signaling pathways and autophagy in the high-risk compared to the low-risk group.

Along the same lines, Wu et al. report a PYR-related lncRNAs prognostic risk signature based on 22 lncRNAs, including AC004847.1, USP30-AS1, AC082651.3, AL033384.1, AC138207.5, AC245041.1, U62317.1, AL512274.1, AC018755.4, MIR200CHG, LINC02362, LINC00861, AL683807.1, AC010503.4, AL512363.1, LINC02437, LINC01527, AL049555.1, AC245041.2, AL365361.1, AC015819.1 and MIR205HG, which stratifies the patients into low- and high-risk groups [104]. The signature predicts OS and correlates with clinical pathological features, including metastasis. ROC and decision-curve analysis indicate that the signature has better diagnostic accuracy than the traditional clinicopathological features. Compared to the low-risk group, the high-risk group showed a reduced proportion of nearly all immune cell subpopulations as well as reduced levels of components of related pathways and functions. In addition, PD-L1 and PD-L2 gene expression levels were lower in the high-risk than in the low-risk group, and differences in the expression of the M6A methylation-related genes (ZC3H13, YTHDF1, FTO, YTHDC2, WTAP) were reported for the two groups. The GSEA analysis showed an enrichment of several pathways, including antigen processing and presentation pathways as well as immune-related pathways in the low-risk group. Drug response prediction analysis indicated different sensitivity to imatinib, isotretinoin, bendamustine, nilotinib, fluphenazine, nelfinavir, oxaliplatin, megestrol acetate, ifosfamide, palbociclib, etoposide, alectinib, and dromostanolone propionate for the two groups.

More recently, Zhong and colleagues reported a 12-PYR-related lncRNAs prognostic signature [105]. Expression levels of PYR-related genes and lncRNA expression data were extracted from TCGA and GTEx. Data from the GEO database were used as a validation set. Differential expression analysis and intersection of mRNA levels with lncRNA expression identified the signature (LINC01234, ZEB1-AS1, SLFN12L, MATN1-AS1, ZNF529-AS1, HOXC-AS2, PLA2G4E-AS1, LRP4-AS1, LINC01028, TM4SF1-AS1, RNF216P1, SNHG17), which was validated by RT-qPCR in A375 and SK-MEL-28 melanoma cell lines. A risk model was constructed and, based on the risk score, TCGA patients were separated into low- and high-risk groups. A significant difference in OS of the two groups was noted following K–M analysis. The nomogram constructed considering the risk model for guiding clinical decision-making showed that the signature efficiently predicted the prognosis. GO and KEGG functional enrichment analysis indicated a significant correlation with immune-related pathways, including regulation of inflammatory response and cytokine production (e.g., IL 1/IL 1β). An enrichment of base-excision repair, glutathione metabolism, oxidative phosphorylation pathways as well as an immune-proficient TME (e.g., DCs, B cells, CD8+ and CD4+ T cells, NK cells, M1 and Treg cells) were observed in the low-risk group. The signature correlated with the levels of autophagy/ferroptosis-related genes as well as with PD1/PD-L1 gene levels. Drug sensitivity prediction analysis indicated differential sensitivity to diastereomer 1, buparlisib, tivozanib, pyrazole anthrone, dasatinib, rapamycin, chelerythrine, JQ1, belinostat, vincristin, methylprednisolone, hydroxyurea, thus highlighting potential therapeutic targets.

## 4. Conclusions

Melanoma is notoriously resistant to anticancer therapies. Numerous genetic, functional and biochemical studies have revealed that melanoma cells become resistant to chemotherapeutics due to their intrinsic resistance to apoptosis and their ability to reprogram their survival as well as proliferation pathways during tumor progression. Intriguing strategies have been employed to counter this, including the discovery of new drugs and the formulation of drug combinations that stimulate non-apoptotic PCD. Indeed, rather than rendering solid tumors immunogenic, available cancer treatments primarily work by inducing non-inflammatory apoptosis or ablation. PYR has received ample attention recently because of its asscociation with innate immunity. This form of PCD, originally thought to be apoptosis due to the similarity between the two pathways (i.e., the caspase dependence), offers an excellent opportunity to alleviate immunosuppression and boost systemic immune responses in the treatment of tumors. PYR effectively modulates the TME, activates a potent T-cell-mediated anti-tumor immune response, inhibits tumor growth and sensitizes cancer cells to chemotherapeutic agents. However, because PYR is an inflammatory mode of cell death, it can also provide an immune microenvironment favorable for tumor growth. Therefore, medical approaches aimed at balancing the two immune-mediated effects (i.e., cancer progression and anticancer potential), although requiring further understanding and investigations, are expected to provide novel opportunities for the treatment of cancer, including melanoma.

The reduction in mortality for melanoma is only in part due to the success of mass-media prevention campaigns. The greatest reduction of mortality is owing to the improvement of checkpoint therapies, and the clinical use of CTLA4 and PD1 inhibitors has benefited many melanoma patients, greatly improving disease-free survival and OS. ICIs suppress tumor immune escape, activate cytotoxic T cells and trigger anti-tumor responses. PYR inducers, or drug combinations that stimulate PYR, can potentiate the antitumor activity of ICIs in both early and advanced melanoma. During early-stage disease, PYR could provide an appropriate immune microenvironment through its proinflammatory effect. As the tumor progresses, its anti-tumor immunity plays a dominant role, and patients affected by an advanced disease are less sensitive to immunotherapy. PYR can stimulate the immune response of these patients by promoting the recruitment of immune cells into the tumor, potentially improving ICI therapy. In addition to melanoma, PYR has been reported in other tumors, including NSCLC, HCC, GC and CRC, and while it is not possible to define a “pyroptotic machinery” specific for melanoma, the presence of a working caspase signaling pathway is suggestive for switching the cell death pathway from non-productive apoptotis to PYR. Various stimulatory signals activate PYR, including PAMPs, DAMPs, drug stimuli and GZM, in turn leading to GSDM activation as well as cell membrane rupture and an inflammatory response. The presence of a working GSDM apparatus represents a salvage condition assigning to GSDMs a crucial role for melanoma cells to undergo PYR instead of apoptosis. Although it is likely that both pathways occur at the same time, PYR represents a potentially potent means that can be harnessed to not only bypass apoptosis resistance but also to activate tumor specific immunity and/or enhance the effectiveness of existing therapies. Therapeutic strategies that increase GSDM levels are intriguing. Among the numerous isoforms, GSDM E is becoming a sort of “Achilles’ heel” for fighting cancer. Though GSDM E rarely emerges in investigations aimed at identifying PYR signatures (Table 2), it appears pivotal for PYR induction following drug treatment (Table 1). 5-FU and lobaplatin induce GSDM E in GC and CRC, respectively, and GSDM E plays a role in the response of melanoma treated with the combinations of BRAFi/MEKi and temozolomide/chloroquine as well. Thus, the induction of GSDM E-mediated PYR may potentially improve treatment outcome and prognosis as well as overcome intrinsic or acquired resistance to apoptosis of patients suffering from melanoma. Additionally, since cancer cells undergoing PYR can evade immune system ‘hunting’ by activating ESCRT-mediated membrane repair, an intriguing strategy to bypass this drawback is the drug combination of PYR inducers with compounds that inhibit ESCRT signaling.

PYR induction for melanoma therapy still requires the elucidation of several issues. Among the studies reported, many need additional investigations and validation. Very often, the in vitro evaluations are incomplete or not sufficiently exhaustive, and in vivo experiments are lacking. Moreover, even if potentially interesting for predicting prognosis and sensitivity to therapeutics, the numerous gene signatures reported need validation. In particular, the studies often lack validations carried out using independent patient cohorts or validations performed on clinical samples. Deeper investigations on clinical samples could help to clarify some inconsistencies found between studies, e.g., the finding that high levels of PD1 and CTLA4 are associated with either a better or worse prognosis from one study to another. A further weakness is the lack of clinical studies that specifically evaluate the induction of PYR in patients suffering from tumors, including melanoma. In this context, the analysis of the expression of GSDMs could be very useful for the definition of their potency as prognostic markers of this PYR-specific family of proteins.

In conclusion, while the recent findings may guide the development of a novel class of therapeutics for the treatment of low immunocompetence melanoma patients, there is a long way to go before we can consider the clinical application of PYR for antitumor therapy in melanoma.

## Figures and Tables

**Figure 1 ijms-24-01285-f001:**
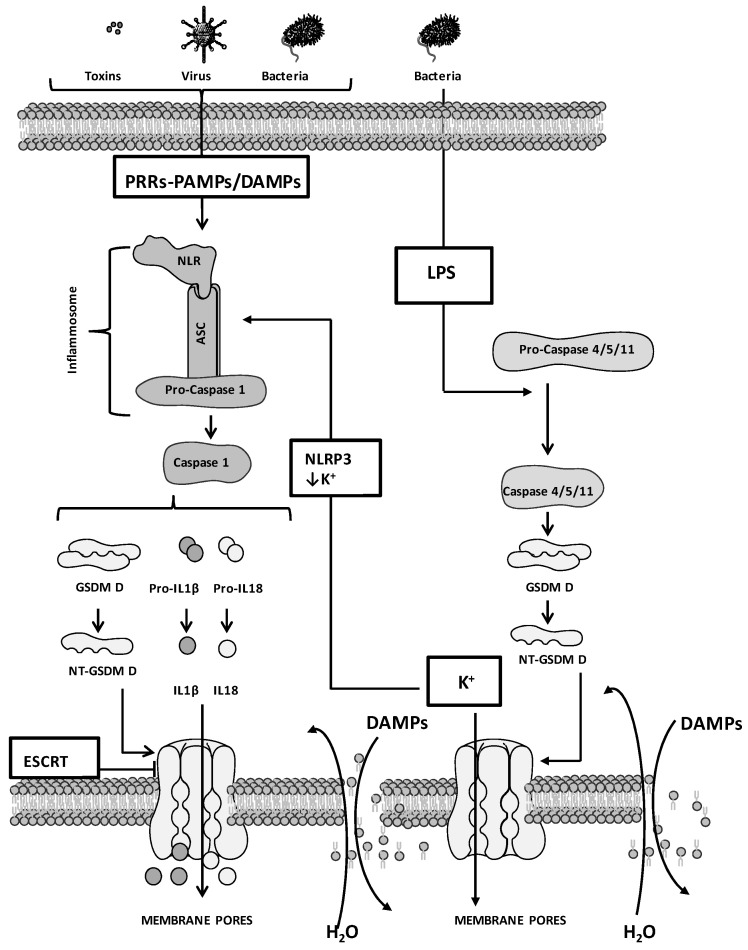
Caspase-dependent mechanisms of pyroptosis. The figure is prepared using tools from Servier Medical Art (http://www.servier.fr/servier-medical-art (accessed on 1 November 2022).

**Figure 2 ijms-24-01285-f002:**
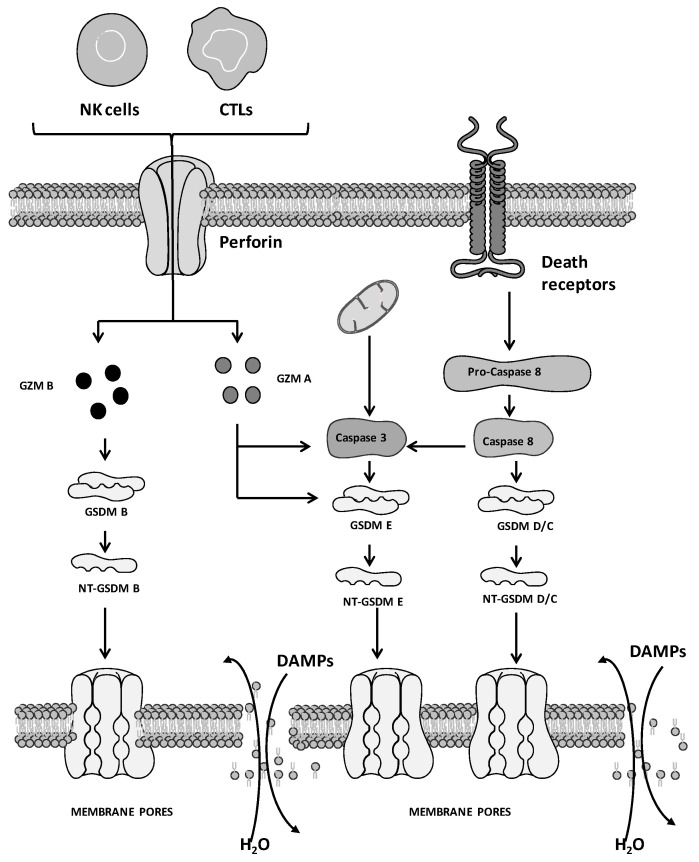
Caspase-independent and caspase 3/8-dependent mechanisms of pyroptosis. The figure is prepared using tools from Servier Medical Art (http://www.servier.fr/servier-medical-art (accessed on 1 November 2022).

**Figure 3 ijms-24-01285-f003:**
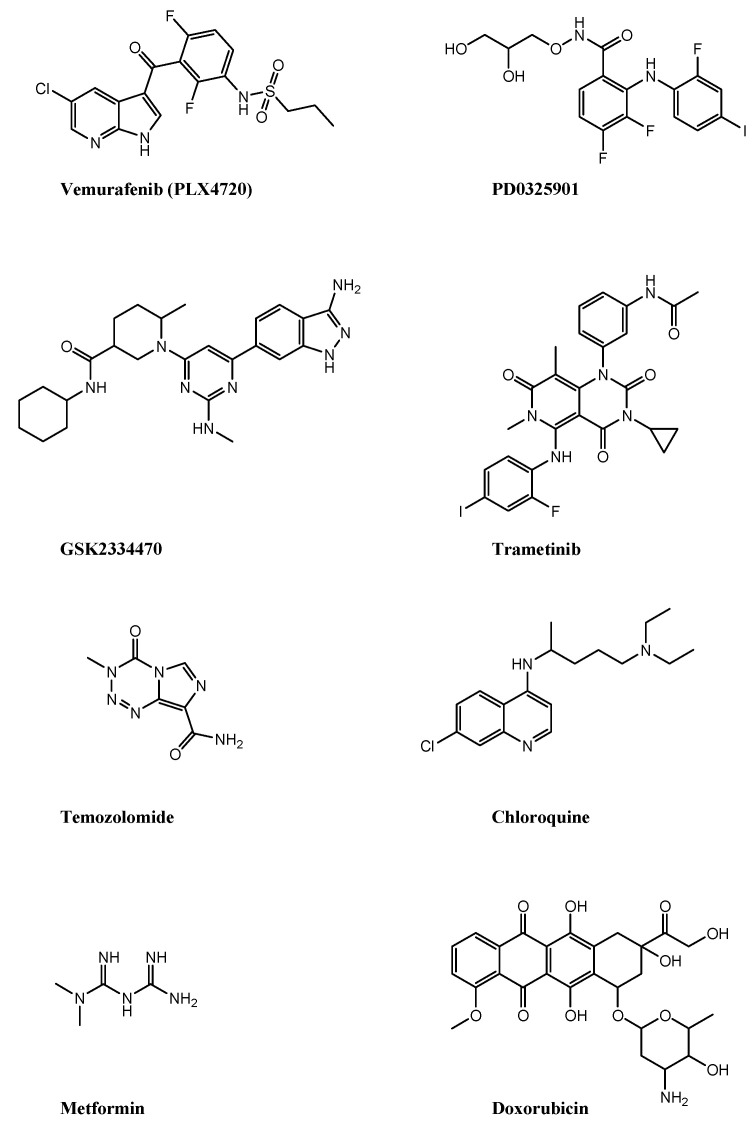
Molecules that induce pyroptosis in melanoma. The chemical structures of BRAF inhibitor, vemurafenib (PLX4720); MEK inhibitors, PD0325901 and trametinib; PDK1 Inhibitor, GSK2334470; DNA alkylating agent, temozolomide; autophagy inhibitor, chloroquine; antihyperglycemic agent, metformin; and toposiomerase II poison, doxorubicin, are reported.

**Table 1 ijms-24-01285-t001:** Drug combinations that induce pyroptosis in melanoma.

Drug Combination	Cell Lines	In Vivo Studies	Pyroptosis Markers
PLX4720/PD0325901	Mouse D4M3.A and YUMM1.7 Human A375 and TJUMEL57	yes	GSDM E HMGB1
GSK2334470/trametinib	Human WM1361A, WM1633, SK-MEL-30 and SK-MEL-173	yes	GSDM E HMGB1
Temozolomide/chloroquine	Human primary culture melanoma cell lines	yes	GSDM E GSDM D
Metformin/doxorubicin	Human A375 and SK-MEL-28	yes	GSDM D

**Table 2 ijms-24-01285-t002:** Gene signatures reflecting pyroptosis in melanoma.

Gene Signature	Cell Line Validation	Drug Sensitivity Prediction	References
GSDM A, GSDM C, IL18, NLRP6, AIM2	Healthy HaCaT and melanoma A375, HS294T and M14	Paclitaxel, docetaxel, cisplatin, sorafenib, PD0325901	[92]
TLR1, CCL8, EMP3, IFNGR2, CCL25, IL15, RTP4, NLRP6	Healthy HaCaT and PIG1 and melanoma A375, SK-MEL-28	Afatinib, sorafenib, refametinib, docetaxel, rapamycin, cisplatin	[93]
GSDM C, GSDM D, NLRP6, IL18, AIM2, PRKACA	No	Anti PD1, anti CTLA4	[94]
GSDM A, GSDM C, AIM2, NOD2	No	Anti PD1, anti CTLA4	[95]
AIM2, IL1B, NLRC4, NLRP3, NLRP6, NLRP7, TNF, ELANE, GSDM A, GSDM B, GSDM C, NLRP1	No	Anti PD1, anti CTLA4	[96]
GSDM C, GZM A, AIM2, PD-L1	No	Nelarabine, dexamethasone decadron, fluphenazine, arsenic trioxide, procarbazine, olaparib, fludarabine, simvastatin, cyclophosphamide, asparaginase	[97]
NLRP9, DHX9, CASP3, NLRC4, AIM2, NLRP3, IL1B, GSDM E, GSDM D	No	No	[98]
CASP5, NEK7, AIM2, CASP1, NLRC4, GSDM D	A375	Anti PD1	[99]
BST2, GBP5, AIM2	No	No	[100]
CASP5, NLRP6, NLRP7, PYCARD	No	Anti PD1, anti CTLA4, adoptive T cell therapy	[101]
IRF9, STAT2	A375 and SK-MEL-28	Increased vemurafenib sensitivity following IRF9 and STAT2 silencing	[102]
AL121603.2, AC107464.2, AC245128.3, AC092171.5, AC242842.1, IRF2-DT, HLA-DQB1-AS1, AC004585.1, LINC00582	No	Bexarotene, bryostatin, docetaxel, bortezomib, bosutinib, camptothecin	[103]
AC004847.1, USP30-AS1, AC082651.3, AL033384.1, AC138207.5, AC245041.1, U62317.1, AL512274.1, AC018755.4, MIR200CHG, LINC02362, LINC00861, AL683807.1, AC010503.4, AL512363.1, LINC02437, LINC01527, AL049555.1, AC245041.2, AL365361.1, AC015819.1, MIR205HG	No	Imatinib, isotretinoin, bendamustine, nilotinib, fluphenazine, nelfinavir, oxaliplatin, megestrol acetate, ifosfamide, palbociclib, etoposide, alectinib, dromostanolone propionate	[104]
LINC01234, ZEB1-AS1, SLFN12L, MATN1-AS1, ZNF529-AS1, HOXC-AS2, PLA2G4E-AS1, LRP4-AS1, LINC01028, TM4SF1-AS1, RNF216P1, SNHG17	A375, SK-MEL-28, PIG1	Diastereomer 1, buparlisib, tivozanib, pyrazole anthrone, dasatinib, rapamycin, chelerythrine, JQ1, belinostat, vincristin, methylprednisolone, hydroxyurea	[105]

## Data Availability

Not applicable.

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
