# Peer review of "The Therapeutic Potential of Pyroptosis in Melanoma"

_ijms, 2023, doi:10.3390/ijms24021285_

Round 1
Reviewer 1 Report
The review is well documented and rather easy to read. The tables are helpful to get a global view on the body of evidence around the main focuses of the manuscript, being the potential drug combinations to increase PYR and the several PYR signatures reported so far and the putative associated drug sensitivity. Nevertheless, the overall quality of the review is somewhat not fully reflected in the conclusion, that is missing real perspectives about how to fill the present gaps in the comprehension and the translation of those findings into tangible benefits for patients, that altogether would be a dramatic added value to the manuscript.

Author Response
Reviewer 1
The review is well documented and rather easy to read. The tables are helpful to get a global view on the body of evidence around the main focuses of the manuscript, being the potential drug combinations to increase PYR and the several PYR signatures reported so far and the putative associated drug sensitivity. Nevertheless, the overall quality of the review is somewhat not fully reflected in the conclusion, that is missing real perspectives about how to fill the present gaps in the comprehension and the translation of those findings into tangible benefits for patients, that altogether would be a dramatic added value to the manuscript.
Authors’ reply. We thank the Reviewer for the positive comments. Manuscript has been edited as suggested. Specifically:
- Line 73 : proteolysi
Authors’ reply: the suggested change has been introduced.
- Line 103 : pyroptotic death.
Authors’ reply: the suggested change has been introduced.
- Line 103 : allowing
Authors’ reply: the suggested change has been introduced.
- Line 165-66 : ,(Figure 2, [71]).
Authors’ reply: the suggested change has been introduced.
- Line 192 : pathogen
Authors’ reply: the suggested change has been introduced.
- Line 204 : oxidative => oxidized forms.
Authors’ reply: the suggested change has been introduced.
- Line 213 : tukonosky => tschonoskii.
Authors’ reply: the suggested change has been introduced.
- Line 212 : please specify colorectal cancer
Authors’ reply: the suggested change has been introduced.
- Line 231 : target
Authors’ reply: the suggested change has been introduced.
- Line 237, please add “in preclinical models” since no validation of the PYR is available in patients with such combinations yet.
Authors’ reply: the suggested change has been introduced.
- Lines 255-56 : chloroquine,; antihyperglycaemic agent, metformin, and toposiomerase II poison, doxorubicin, are reported.
Authors’ reply: the suggested changes have been introduced.
- Line 267 : the combination GSK2334470/trametinib
Authors’ reply: the suggested change has been introduced.
- Line 326 : apply
Authors’ reply: the suggested change has been introduced.
- Line 328 : I understand you consider BRAFi as a class of PYR inducers, but sorafenib can hardly be considered a BRAFi while it is much more potent against VEGFRs, PDGFRs, KIT and FLT3. Please adjust your assertion.
Authors’ reply: we agree with the Reviewer and we have corrected the manuscript accordingly.
- Line 349 : Compared to high risk ones, low risk ...
Authors’ reply: the suggested change has been introduced.
- Line 353 : Please provide a bit more details about the outcome of this study.
Authors’ reply: the suggested we have implemented the manuscript introducing the limitations of the study underlined by the authors. Specifically we have detailed “Though the study underlines that PYR and inflammation responses predict prognosis and immunotherapy response of patients suffering from melanoma, the authors evidence two major limitations, including i) the lack of an independent patient cohort to better validate the prognostic power of the model and ii) the lack of validation resulting from the analysis of clinical samples.”
Line 356 : Quite intriguing that from one study to the other, the PD-1 and CTLA4 high expressing groups are either of better or worse prognostic. any potential explanation? Authors’ reply: we agree with the Reviewer. Although it is difficult to clarify this issue, we have partially faced it into the conclusions.
- Line 389 : ICI
Authors’ reply: the suggested change has been introduced.
- Line 402 : benefit
Authors’ reply: the suggested change has been introduced.
- Line 418 : nine-gene
Authors’ reply: the suggested change has been introduced.
- Line 431 : Unclear why there are 17 genes initially and then only 6?
Authors’ reply: we have clarified this point. Specifically we have introduced: “In order to improve the model accuracy and decrease model overfitting, LASSO analysis is applied and a more accurate risk model based on the expression of 6 PYR-associated genes (CASP5, NEK7, AIM2, CASP1, NLRC4, GSDM D) is defined.”
- Line 437 : GSDMD => GSDM D.
Authors’ reply: the suggested change has been introduced.
- Line 475 : clear cell renal cell
Authors’ reply: the suggested change has been introduced.
- Line 475 : KEEG => KEGG.
Authors’ reply: the suggested change has been introduced.
- Line 483 : polarized macrophages M1 and M2
Authors’ reply: the suggested change has been introduced.
- Line 484 : Please harmonize “CD4+ T cells” across the manuscript.
Authors’ reply: we have harmonized the manuscript accordingly.
- Line 551 : what distinguishes CD8+ from cytotocix T cells? Same comment as for line 484. Authors’ reply: we have harmonized the manuscript accordingly.
- Table 2 : HaCa T => HaCaT (x2).
Authors’ reply: the suggested changes have been introduced.
- Line 568 : immune response
Authors’ reply: the suggested change has been introduced.
- Lines 574-77 : the sentence does not bring anything to the argumentation. I would recommend to delete it.
Authors’ reply: we have deleted the sentence as suggested.
- Line 588 : improve
Authors’ reply: the suggested change has been introduced.
- Line 588 : as to
Authors’ reply: the suggested change has been introduced.
- Lines 593 to 601 : Please restructure the conclusion to separate more clearly what is related to the clinical relevance of PYR from what is related to the surrogate markers of PYR in cancer prognosis. It may also be interesting to devise a bit more about the potential added value of those PYR signatures over PD-L1 protein expression to select patients.
Authors’ reply: we agree with the Reviewer and we have edited the conclusions accordingly.
- Finally, it may be much valuable for the community if you could propose potential ways to further validate the preclinical and in silico findings.
Authors’ reply: we agree with the Reviewer and we have discussed this point in the conclusions.

Reviewer 2 Report
Pyroptosis has received ample attention recently because of its association with innate immunity. This form of programmed cell death was originally thought to be apoptosis because some of its characteristics were similar to apoptosis, such as caspase-dependent. Numerous reviews on pyroptosis related to cancer have been reported. In this review paper, the authors provide an assessment of pyroptosis therapeutic potential specifically in melanoma. However, there are a few concerns. It is not clear why pyroptosis is unique to melanoma. Is there any special machinery specific to melanoma? Is it possible to evaluate what is the deciding factor for melanoma cells to undergo pyroptosis instead of apoptosis? Are both pathways occurring at the same time or superseding each other?
The author mentioned that the reduction in mortality is due to the success of mass media prevention campaigns as well as medical management. It is absolutely accurate; nevertheless, it is noteworthy that the greatest reduction of mortality in melanoma is owing to the improvement of checkpoint therapies (i.e. anti-PD1). Thus, the extensive review geared toward pyroptosis and immune alteration in melanoma may help strengthen the overall rationale.
The author did a wonderful job breaking down the important information in the Table format, however, the references are not clear. For suggestions, in Table 2, an additional column with listed references may be included alongside Gene signature, Cell line validation, and Drug sensitivity prediction.
Overall, this is a well-written review with a comprehensive evaluation of the pyroptosis mechanism which can be further improved by emphasizing more on the importance of pyroptosis in the melanoma setting.
Author Response
Reviewer 2
Pyroptosis has received ample attention recently because of its association with innate immunity. This form of programmed cell death was originally thought to be apoptosis because some of its characteristics were similar to apoptosis, such as caspase-dependent. Numerous reviews on pyroptosis related to cancer have been reported. In this review paper, the authors provide an assessment of pyroptosis therapeutic potential specifically in melanoma. However, there are a few concerns. It is not clear why pyroptosis is unique to melanoma. Is there any special machinery specific to melanoma? Is it possible to evaluate what is the deciding factor for melanoma cells to undergo pyroptosis instead of apoptosis? Are both pathways occurring at the same time or superseding each other?
Authors’ reply. We thank the Reviewer for the comments. We have discussed the relevance of pyroptosis for melanoma in the conclusions. The manuscript has been edited as suggested. Specifically:
- The author mentioned that the reduction in mortality is due to the success of mass media prevention campaigns as well as medical management. It is absolutely accurate; nevertheless, it is noteworthy that the greatest reduction of mortality in melanoma is owing to the improvement of checkpoint therapies (i.e. anti-PD1). Thus, the extensive review geared toward pyroptosis and immune alteration in melanoma may help strengthen the overall rationale.
Authors’ reply: we agree with the Reviewer and we have discussed this point in the conclusions.
- The author did a wonderful job breaking down the important information in the Table format, however, the references are not clear. For suggestions, in Table 2, an additional column with listed references may be included alongside Gene signature, Cell line validation, and Drug sensitivity prediction.
Authors’ reply: table 2 has been changed as suggested.
- Overall, this is a well-written review with a comprehensive evaluation of the pyroptosis mechanism which can be further improved by emphasizing more on the importance of pyroptosis in the melanoma setting.
Authors’ reply: we agree with the Reviewer and we have discussed this point in the conclusions.
